# Verifiably Robust Conformal Prediction

**Linus Jeary**[*]
Department of Informatics
King's College London, UK
linus.jeary@kcl.ac.uk

**Tom Kuipers**[*]
Department of Informatics
King's College London, UK
tom.kuipers@kcl.ac.uk

**Mehran Hosseini**
Department of Informatics
King's College London, UK
mehran.hosseini@kcl.ac.uk

**Nicola Paoletti**
Department of Informatics
King's College London, UK
nicola.paoletti@kcl.ac.uk

## Abstract

Conformal Prediction (CP) is a popular uncertainty quantification method that provides distribution-free, statistically valid prediction sets, assuming that training and test data are exchangeable. In such a case, CP's prediction sets are guaranteed to cover the (unknown) true test output with a user-specified probability. Nevertheless, this guarantee is violated when the data is subjected to adversarial attacks, which often result in a significant loss of coverage. Recently, several approaches have been put forward to recover CP guarantees in this setting. These approaches leverage variations of randomised smoothing to produce conservative sets which account for the effect of the adversarial perturbations. They are, however, limited in that they only support $\ell_2$-bounded perturbations and classification tasks. This paper introduces *VRCP (Verifiably Robust Conformal Prediction)*, a new framework that leverages recent neural network verification methods to recover coverage guarantees under adversarial attacks. Our VRCP method is the first to support perturbations bounded by arbitrary norms including $\ell_1$, $\ell_2$, and $\ell_\infty$, as well as regression tasks. We evaluate and compare our approach on image classification tasks (CIFAR10, CIFAR100, and TinyImageNet) and regression tasks for deep reinforcement learning environments. In every case, VRCP achieves above nominal coverage and yields significantly more efficient and informative prediction regions than the SotA.

## 1 Introduction

Conformal Prediction (CP) (Vovk et al., 2005; Angelopoulos and Bates, 2021) is a popular uncertainty quantification method. In essence, it is a model-agnostic, distribution-free framework that allows one to construct prediction sets that are guaranteed to include the true (unknown) output with probability greater than $1 - \alpha$, where $\alpha \in (0, 1)$ is a user-specified miscoverage/error rate. In other words, for a test point $(\boldsymbol{x}_{n+1}, y_{n+1})$, CP seeks to construct a prediction set $C(\boldsymbol{x}_{n+1})$ such that the following coverage (a.k.a. validity) guarantee holds:

$$\mathbb{P}_{\boldsymbol{x}_{n+1}, y_{n+1}}[y_{n+1} \in C(\boldsymbol{x}_{n+1})] \geq 1 - \alpha. \tag{1}$$

Importantly, the above guarantee holds when the calibration data, used to construct $C(\boldsymbol{x}_{n+1})$, and the test point are exchangeable (a special case is when calibration and test data are i.i.d.).

---

[*]Authors contributed equally.

38th Conference on Neural Information Processing Systems (NeurIPS 2024).

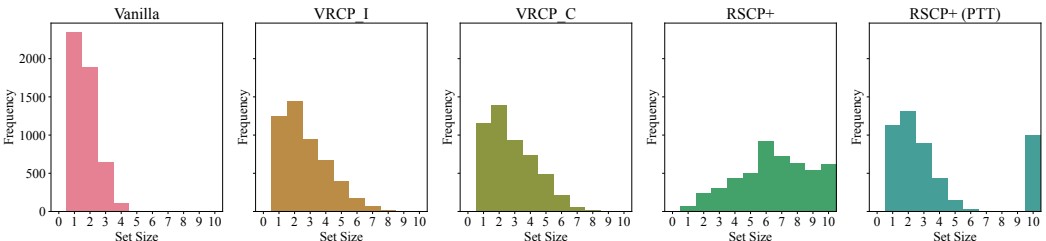

Figure 1: Distribution of prediction set sizes for vanilla conformal prediction (vanilla CP) which violates Eq. (2), as well as for our proposed robust algorithms (VRCP–I and VRCP–C) along with the SotA (RSCP+ and RSCP+ (PTT), see Section 3) on the CIFAR10 dataset. As we observe, VRCP–I and VRCP–C closely resemble the spread of vanilla CP prediction set sizes, whilst the SotA falls short of achieving this. Here we use an adversarial perturbation of radius $\epsilon = 0.02$, error rate $\alpha = 0.1$, number of splits $n_{\text{splits}} = 50$ and smoothing parameter (used in RSCP+ and RSCP+ (PTT)) $\sigma = 2\epsilon$.

When exchangeability is violated, e.g., in the presence of test-time distribution shifts, CP's coverage guarantee (1) ceases to hold, and we cannot rely on the prediction sets it produces. In this work, we address shifts induced by adversarial perturbations on the test inputs. In particular, we focus on perturbations in the form of additive $\ell_p$-bounded noise.

To recover guarantees under adversarial inputs, the general mechanism is to inflate the prediction set to permit larger degrees of uncertainty. However, special care must be taken to avoid producing overly large or even trivial sets – i.e. those containing all possible outputs – as such sets do not provide any useful inference.

**Contributions** We propose a CP framework that provides statistically valid prediction sets despite the presence of $\ell_p$-bounded adversarial perturbations at inference time. Formally, for any adversarially perturbed test point $\tilde{\boldsymbol{x}}_{n+1} = \boldsymbol{x}_{n+1} + \boldsymbol{\delta}$, our method produces adversarially robust sets $C_\epsilon$ that enjoy the following guarantee:

$$\mathbb{P}[y_{n+1} \in C_\epsilon(\tilde{\boldsymbol{x}}_{n+1})] \geq 1 - \alpha \quad \forall \boldsymbol{\delta} \text{ s.t. } \|\boldsymbol{\delta}\|_p \leq \epsilon. \tag{2}$$

While CP uses an underlying predictor $f$, often a neural network (NN), to construct prediction regions, the novelty of our approach is to leverage NN verification algorithms to compute upper and lower output bounds of $f(\boldsymbol{x}')$ for any $\boldsymbol{x}' \in B_\epsilon(\boldsymbol{x}) = \{\boldsymbol{x}' : \|\boldsymbol{x}' - \boldsymbol{x}\|_p \leq \epsilon\}$. We use these bounds to inflate the CP regions, resulting in provably robust and *efficient* prediction sets. To the best of our knowledge, this is the first work that combines NN verification algorithms and CP to construct adversarially robust prediction sets. We call our method *VRCP (Verifiably Robust Conformal Prediction)*.

Recent work (discussed in Section 3) achieves adversarially robust coverage using probabilistic methods, specifically, randomised smoothing (Cohen et al., 2019). Our approach overcomes some of the theoretical and empirical drawbacks of these prior methods, which are restricted to classification tasks with $\ell_2$-norm bounded guarantees and are overly conservative in practice.

Thanks to our verification-based approach, VRCP is the first to extend adversarially robust conformal prediction to regression tasks and the first to go beyond $\ell_2$-norm bounded guarantees. In Section 4, we introduce two versions of VRCP that apply verification at calibration and inference time, respectively. Further, in Section 5, we empirically validate our theoretical guarantees and demonstrate a direct improvement over previous work in terms of prediction set efficiency (i.e., average set size) compared to prior work. Fig. 1 shows an extract of our results on CIFAR10, demonstrating that VRCP yields more informative (tighter) prediction regions, a trend that we observe experimentally across all our benchmarks.

## 2 Preliminaries

We denote with $\mathbb{R}_+$ the set of positive real numbers. Vectors $\boldsymbol{x} \in \mathbb{R}^d$ are shown in bold italic and scalars $x \in \mathbb{R}$ in italic typeface. We denote the norm used to make $\mathbb{R}^d$ a normed vector space by $\|\cdot\|$. This could for instance be $\ell_1, \ell_2$, or $\ell_\infty$-norm. Whenever a specific norm is intended, we indicate it using an index, e.g., $\|\cdot\|_2$ indicates the $\ell_2$-norm. We denote the $\epsilon$-ball around a point $\boldsymbol{x} \in \mathbb{R}^d$ with respect to the used norm by $B_\epsilon(\boldsymbol{x})$.

## 2.1 Conformal Prediction

We provide a brief overview of the inductive (or split) vanilla CP approach. Suppose we have a dataset $\mathcal{D}$ containing pairs $(\boldsymbol{x}, y)$ sampled i.i.d. from an (unknown) data-generating distribution over a feature space $X \subseteq \mathbb{R}^d$ and label space $Y$ such that $\mathcal{D} = \{(\boldsymbol{x}_1, y_1), \ldots, (\boldsymbol{x}_m, y_m)\}$.

We partition the dataset into disjoint training and calibration sets $\mathcal{D}_{\text{train}}$ and $\mathcal{D}_{\text{cal}}$, letting $n = |\mathcal{D}_{\text{cal}}|$. We fit a predictor $f$ on $\mathcal{D}_{\text{train}}$ and define a score function $S : (X \times Y) \to \mathbb{R}$ as some notion of prediction error, such as $S(\boldsymbol{x}, y) = \|f(\boldsymbol{x}) - y\|$ when $f$ is a regressor, or $S(\boldsymbol{x}, y) = 1 - f(\boldsymbol{x})_y$ when $f$ is a classifier with $f(\cdot)_y$ being $y$'s predicted likelihood.

After applying the score function to all calibration points, we construct the score distribution as $F = {\delta_\infty}/{(n+1)} + \sum_{i=1}^n {\delta_{s_i}}/{n+1}$, where $\delta_s$ is the Dirac distribution with parameter $s$, $s_i = S(\boldsymbol{x}_i, y_i)$ and $\delta_\infty$ represents the unknown score (potentially infinite) of the test point.

Given a miscoverage/error rate $\alpha$ and a test point $(\boldsymbol{x}_{n+1}, y_{n+1})$, we define the prediction set $C(\boldsymbol{x}_{n+1})$ by including all labels that appear sufficiently likely w.r.t. the score distribution: $C(\boldsymbol{x}_{n+1}) = \{y \in Y : S(\boldsymbol{x}_{n+1}, y) \leq Q_{1-\alpha}(F)\}$, where $Q_{1-\alpha}(F)$ is the $1 - \alpha$ quantile of $F$. This set satisfies the marginal coverage guarantee in Eq. (1) if the test point and the calibration points are exchangeable.

## 2.2 Adversarial Attacks

Neural networks have been shown to be vulnerable to *adversarial attacks*, i.e., small changes to their input that jeopardise the prediction (Szegedy et al., 2014; Biggio and Roli, 2018). This notion can be formally defined as maximising an adversarial objective function (e.g., the loss of the true label) subject to $\|\boldsymbol{x} - \tilde{\boldsymbol{x}}\| \leq \epsilon$. Alternatively, it can be defined as finding an adversarial example $\tilde{\boldsymbol{x}} \in \mathbb{R}^m$, such that $\|\boldsymbol{x} - \tilde{\boldsymbol{x}}\| \leq \epsilon$ and $\|f(\tilde{\boldsymbol{x}}) - y\| \geq \delta$ for a given neural network $f : \mathbb{R}^d \to \mathbb{R}^n$.

## 2.3 Neural Network Verification

Various approaches have been proposed to verify the robustness of NNs against adversarial attacks. These approaches can be divided into complete and incomplete algorithms. Given a neural network $f$, a verifier is *complete* if it allows computing exact bounds $f^\perp$ and $f^\top$ for the image $f(B_\epsilon(\boldsymbol{x})) = \{f(\boldsymbol{x}') : \boldsymbol{x}' \in B_\epsilon(\boldsymbol{x})\}$, i.e., such that

$$f^\perp = \min_{\boldsymbol{x}' \in B_\epsilon(\boldsymbol{x})} \{f(\boldsymbol{x}')\}, \quad f^\top = \max_{\boldsymbol{x}' \in B_\epsilon(\boldsymbol{x})} \{f(\boldsymbol{x}')\}, \tag{3}$$

where $\min$ and $\max$ are computed coordinate-wise for vector-valued NNs. A verifier is *incomplete*, but *sound*, if it computes bounds that are valid but not exact, i.e., such that:

$$f^\perp \leq \min_{\boldsymbol{x}' \in B_\epsilon(\boldsymbol{x})} \{f(\boldsymbol{x}')\}, \quad f^\top \geq \max_{\boldsymbol{x}' \in B_\epsilon(\boldsymbol{x})} \{f(\boldsymbol{x}')\}. \tag{4}$$

Our results are verifier-agnostic, meaning that they are valid for any verifier that can produce exact bounds (as in Eq. (3)) or conservative bounds (as in Eq. (4)), depending on the completeness or incompleteness of the verifier used. The fastest and simplest way to compute the bounds in Eq. (4) is to propagate the bounds on the input $B_\epsilon(\boldsymbol{x})$ through the network to compute the output bounds. Several methods based on this approach have been proposed (Gowal et al., 2018; Wang et al., 2018; Zhang et al., 2018a; Batten et al., 2024; Lopez et al., 2023). At the expense of fast computation speed, these methods may result in loose bounds in Eq. (4). On the other hand, several complete methods (Pulina and Tacchella, 2010; Katz et al., 2017; Hosseini and Lomuscio, 2023) for NN verification have been put forward. Even though these methods compute exact bounds, their downside is their high computational cost.

## 3 Related Work

**Adversarially Robust Conformal Prediction** Gendler et al. (2021) introduced an algorithm called Randomly Smoothed Conformal Prediction (RSCP) that integrates randomised smoothing (Duchi et al., 2012; Cohen et al., 2019; Salman et al., 2019) with CP to provide robust coverage under adversarial attacks. RSCP replaces the CP score function $S(\boldsymbol{x}, y)$ with a smoothed score $\widetilde{S}(\boldsymbol{x}, y)$ obtained by averaging the values of $S(\boldsymbol{x} + \boldsymbol{v}, y)$ over $n_{\text{MC}}$ realisations of a Gaussian noise

vector $\boldsymbol{v} \sim \mathcal{N}(0, \sigma^2 I)$, for a given smoothing level $\sigma$. To correct for potential $\ell_2$-norm bounded $\epsilon$ perturbations at inference time, the critical value computed over the smoothed scores is inflated by $\epsilon/\sigma$. This method produces empirically sound results, but the provided formal guarantees were found to be invalid in a later work (Yan et al., 2023), as discussed below.

**Provably Robust Conformal Prediction**  Yan et al. (2023) address the issue with the robustness guarantee of Gendler et al. (2021) by correctly bounding the estimation error caused by the Monte-Carlo sampling used when generating the smoothed scores. The bound introduces an additional hyperparameter $\beta$ such that they now find the $Q_{1-\alpha+2\beta}$ of smooth calibrated scores and inflate by Hoeffding's bound $\sqrt{-\ln(\beta)/2n_{\mathrm{MC}}}$ before correcting by $\epsilon/\sigma$. Furthermore, the smoothed scores of the test points are decreased by an empirical Bernstein bound. This further inflation of the critical value and deflation of smooth scores for each test point often cause their amended algorithm, so-called RSCP+, to generate trivial prediction sets.

To address this issue, the authors introduce two methods to improve the efficiency of RSCP+. Firstly, they use robustly calibrated training (RCT), a training-time regularisation technique that penalises NN parameters that contribute to high scores for the true label. Our approach assumes that the underlying classifier is given; hence, we do not evaluate RCT in our experiments.

Secondly, they implement a post-training transformation (PTT), which aims to decrease the values of the smoothed calibration scores that lie between $Q_{1-\alpha}$ (the critical value of the base scores) and $\widetilde{Q}_{1-\alpha}$ (that of the smoothed scores). To this purpose, they transform the CDF of the smoothed scores $\widetilde{S}$ by composing learned ranking and sigmoid transformations with hyperparameters $b$ and $T$ using a holdout set $\mathcal{D}_{\mathrm{hold}}$ sampled i.i.d from $\mathcal{D}_{\mathrm{cal}}$. PTT however is not theoretically guaranteed to improve the average set sizes computed by RSCP+ and, in many cases, its efficacy is largely dependent on how representative the sampled holdout set is of the calibration set. We demonstrate the effect of PTT's sample dependence empirically in Section 5.1.

**Probabilistically Robust Conformal Prediction**  Ghosh et al. (2023) also focus on the adversarial setting but maintain a relaxed form of robust coverage, where input perturbations $\boldsymbol{\delta}$ are drawn from a specific distribution and only a proportion of such perturbations are sought to be covered. In contrast, we do not make assumptions about the noise distribution, and we account for any $\epsilon$-bounded perturbation.

All the works[2] discussed here rely on randomised smoothing Duchi et al. (2012) and as such are limited to the $\ell_2$-norm. In contrast, our VRCP approach relies on NN verifiers, can be used with any $\ell_p$-norms supported by the verification method, and does not require smoothing hyperparameters or holdout sets.

## 4  Verifiably Robust Conformal Prediction (VRCP)

In this section, we formally introduce two variants of VRCP. Both methods allow us to construct adversarially robust prediction sets at inference time.

The first variant, *VRCP via Robust Inference (VRCP–I)*, employs NN verification at inference time to compute a lower bound of the score for the given test input (best-case score), thereby obtaining more conservative regions. The calibration procedure is computed as in standard CP. The second variant, *VRCP via Robust Calibration (VRCP–C)*, instead uses NN verification at calibration time to derive upper bounds for the calibration scores (worst-case), thereby obtaining a more conservative calibration threshold (critical value). This allows us to use the regular scores at inference time, without requiring NN verification.

### 4.1  Verifiably Robust Conformal Prediction via Robust Inference (VRCP–I)

Given a calibration set $\mathcal{D}_{\mathrm{cal}}$, a test input $\boldsymbol{x}_{n+1}$, and score function $S(\cdot, \cdot)$, we compute the prediction set for $\boldsymbol{x}_{n+1}$ as follows.

---

[2]We are aware of related contemporaneous work by Zargarbashi et al. (2024). However, at the time of submission, neither the manuscript nor the code were available.

1. For each $y \in Y$ we compute,

$$s^{\perp}(\boldsymbol{x}_{n+1}, y) \leq \inf_{\boldsymbol{x}' \in B_{\epsilon}(\boldsymbol{x}_{n+1})} S(\boldsymbol{x}', y). \tag{5}$$

2. The robust prediction set is then defined as

$$C_{\epsilon}(\boldsymbol{x}_{n+1}) = \{y : s^{\perp}(\boldsymbol{x}_{n+1}, y) \leq Q_{1-\alpha}(F)\}. \tag{6}$$

Below, we show that we are able to maintain the marginal coverage guarantee from Eq. (2) for any $\ell_p$-norm bounded adversarial attack.

**Theorem 1.** *Let $\tilde{\boldsymbol{x}}_{n+1} = \boldsymbol{x}_{n+1} + \boldsymbol{\delta}$ for a clean test sample $\boldsymbol{x}_{n+1}$ and $\|\boldsymbol{\delta}\|_p \leq \epsilon$. The prediction set $C_{\epsilon}(\tilde{\boldsymbol{x}}_{n+1})$ defined in Eq. (6) satisfies $\mathbb{P}\left[y_{n+1} \in C_{\epsilon}(\tilde{\boldsymbol{x}}_{n+1})\right] \geq 1 - \alpha$.*

*Proof.* We obtain that

$$
\begin{aligned}
\mathbb{P}\left[y_{n+1} \in C_{\epsilon}(\tilde{\boldsymbol{x}}_{n+1})\right] &= \mathbb{P}\left[s^{\perp}(\tilde{\boldsymbol{x}}_{n+1}, y_{n+1}) \leq Q_{1-\alpha}(F)\right] \\
&\geq \mathbb{P}\left[\inf_{\boldsymbol{x}' \in B_{\epsilon}(\tilde{\boldsymbol{x}}_{n+1})} S(\boldsymbol{x}', y_{n+1}) \leq Q_{1-\alpha}(F)\right] \qquad \text{by Eq. (5)} \\
&\geq \mathbb{P}\left[S(\boldsymbol{x}_{n+1}, y_{n+1}) \leq Q_{1-\alpha}(F)\right] \geq 1 - \alpha. \qquad \square
\end{aligned}
$$

## 4.2  Verifiably Robust Conformal Prediction via Robust Calibration

Given a calibration set $\mathcal{D}_{\text{cal}}$, a test input $\boldsymbol{x}_{n+1}$, and score function $S(\cdot, \cdot)$, we compute the robustly calibrated prediction set for $\boldsymbol{x}_{n+1}$ as follows.

1. We compute the upper-bound calibration distribution as:

$$F^{\top} = \frac{\delta_{\infty}}{(n+1)} + \sum_{i=1}^{n} \frac{\delta_{s_i^{\top}}}{n+1}, \text{ where } s_i^{\top} \geq \sup_{\boldsymbol{x}' \in B_{\epsilon}(\boldsymbol{x}_i)} S(\boldsymbol{x}', y_i). \tag{7}$$

2. The robust post-calibration prediction set is then defined as

$$C_{\epsilon}(\boldsymbol{x}_{n+1}) = \{y : S(\boldsymbol{x}_{n+1}, y) \leq Q_{1-\alpha}(F^{\top})\}. \tag{8}$$

**Theorem 2.** *Let $\tilde{\boldsymbol{x}}_{n+1} = \boldsymbol{x}_{n+1} + \boldsymbol{\delta}$ for a clean test sample $\boldsymbol{x}_{n+1}$ and $\|\boldsymbol{\delta}\|_p \leq \epsilon$. The prediction set $C_{\epsilon}(\tilde{\boldsymbol{x}}_{n+1})$ defined in Eq. (8) satisfies $\mathbb{P}\left[y_{n+1} \in C_{\epsilon}(\tilde{\boldsymbol{x}}_{n+1})\right] \geq 1 - \alpha$.*

*Proof.* We have that

$$
\begin{aligned}
\mathbb{P}\left[y_{n+1} \in C_{\epsilon}(\tilde{\boldsymbol{x}}_{n+1})\right] &= \mathbb{P}\left[S(\tilde{\boldsymbol{x}}_{n+1}, y_{n+1}) \leq Q_{1-\alpha}\left(F^{\top}\right)\right] \\
&\geq \mathbb{P}\left[S(\tilde{\boldsymbol{x}}_{n+1}, y_{n+1}) \leq Q_{1-\alpha}\left(\left\{\sup_{\boldsymbol{x}' \in B_{\epsilon}(\boldsymbol{x}_i)} S(\boldsymbol{x}', y_i)\right\}_{i=1}^{n} \cup \{\infty\}\right)\right] \\
&\geq \mathbb{P}\left[\sup_{\boldsymbol{x}' \in B_{\epsilon}(\boldsymbol{x}_{n+1})} S(\boldsymbol{x}', y_{n+1}) \leq Q_{1-\alpha}\left(\left\{\sup_{\boldsymbol{x}' \in B_{\epsilon}(\boldsymbol{x}_i)} S(\boldsymbol{x}', y_i)\right\}_{i=1}^{n} \cup \{\infty\}\right)\right] \\
&\geq 1 - \alpha
\end{aligned}
$$

Let $P^{\top}$ denote the distribution of $(\boldsymbol{x}^{\top}, y)$ where $\boldsymbol{x}^{\top} = \text{argsup}_{\boldsymbol{x}' \in B_{\epsilon}(\boldsymbol{x})} S(\boldsymbol{x}', y)$ and $(\boldsymbol{x}, y) \sim P$. The final inequality above holds since it is equivalent to constructing a CP prediction set using $n$ i.i.d realisations of $P^{\top}$ and evaluating it on $\boldsymbol{x}_{n+1} \sim P^{\top}$. The resulting set will include the true test output $y_{n+1}$ with probability at least $1 - \alpha$. $\qquad \square$

## 4.3  Computation of score bounds

**Classification**   In the classification setting, we use the score function proposed in (Lei et al., 2013; Gendler et al., 2021):

$$S(\boldsymbol{x}, y) = 1 - f(\boldsymbol{x})_y, \tag{9}$$

where $f(\boldsymbol{x})_y \in (0,1)$ is the model-predicted likelihood for label $y$. In this setting, to compute $s^\perp$ and $s^\top$ (required by VRCP–I and VRCP–C, respectively), it suffices to use NN verification algorithms to derive the output bounds for $f(\boldsymbol{x})$. Specifically, in VRCP–I, for a test input $\boldsymbol{x}_{n+1}$ and for each $y \in Y$ we derive $s^\perp(\boldsymbol{x}_{n+1}, y)$ as

$$s^\perp(\boldsymbol{x}_{n+1}, y) = 1 - f(\boldsymbol{x}_{n+1})_y^\top, \tag{10}$$

where $f(\boldsymbol{x}_{n+1})_y^\top$ denotes the upper bound computed by the neural network verifier for the model-predicted likelihood of label $y \in Y$ and input $\boldsymbol{x}_{n+1}$.

In VRCP–C, for each calibration point $(\boldsymbol{x}_i, y_i)$ we compute $s^\top(\boldsymbol{x}_i, y_i)$ as

$$s^\top(\boldsymbol{x}_i, y_i) = 1 - f(\boldsymbol{x}_i)_{y_i}^\perp, \tag{11}$$

where $f(\boldsymbol{x}_i)_{y_i}^\perp$ denotes the lower bound of the model output for label $y_i$ given input $\boldsymbol{x}_i$.

**Regression**  In the regression tasks, we follow the conformalized quantile regression (CQR) methodology proposed by (Romano et al., 2019). We train quantile regressors $f_{\alpha_{\text{low}}}$ and $f_{\alpha_{\text{high}}}$ to estimate the $\alpha_{\text{low}} = \alpha/2$ and $\alpha_{\text{high}} = 1 - \alpha/2$ quantiles of $y \mid \boldsymbol{x}$. In CQR, we use the following score function:

$$S(\boldsymbol{x}, y) = \max\{f_{\alpha_{\text{low}}}(\boldsymbol{x}) - y, y - f_{\alpha_{\text{high}}}(\boldsymbol{x})\}. \tag{12}$$

Unlike classification, where the label space is discrete, we cannot construct the region explicitly by enumerating all possible outputs $y$. Instead, the prediction region for a given test point $C(\boldsymbol{x}_{n+1})$ is constructed implicitly, by adjusting the quantile predictions by the critical value of the calibration distribution $Q_{1-\alpha}(F)$, as follows:

$$C(\boldsymbol{x}_{n+1}) = \left[f_{\alpha_{\text{low}}}(\boldsymbol{x}_{n+1}) - Q_{1-\alpha}(F), f_{\alpha_{\text{high}}}(\boldsymbol{x}_{n+1}) + Q_{1-\alpha}(F)\right] \tag{13}$$

In both VRCP–C and VRCP–I, the score function leverages an NN verifier to derive the bounds over the upper and lower quantiles of the model. In VRCP–C, we compute the worst-case calibration scores as:

$$s^\top(\boldsymbol{x}_i, y_i) = \max\{f_{\alpha_{\text{low}}}^\top(\boldsymbol{x}_i) - y_i, y_i - f_{\alpha_{\text{high}}}^\perp(\boldsymbol{x}_i)\}. \tag{14}$$

In VRCP–I for classification, for each output we check inclusion in $C_\epsilon$ by using the best-case score $s^\perp$. As explained above, explicit enumeration is infeasible for regression, and so we construct our robust region by replacing predicted quantiles in Eq. (13) with their conservative approximations, as follows:

$$C_\epsilon(\boldsymbol{x}_{n+1}) = \left[f_{\alpha_{\text{low}}}^\perp(\boldsymbol{x}_{n+1}) - Q_{1-\alpha}(F), f_{\alpha_{\text{high}}}^\top(\boldsymbol{x}_{n+1}) + Q_{1-\alpha}(F)\right] \tag{15}$$

The above-defined region is equivalent to enumerating all possible outputs $y$, and for each, considering the best-case score $s^\perp(\boldsymbol{x}_{n+1}, y) = \max\{f_{\alpha_{\text{low}}}^\perp(\boldsymbol{x}_{n+1}) - y, y - f_{\alpha_{\text{high}}}^\top(\boldsymbol{x}_{n+1})\}$. A proof is available in Appendix A.

A nice property of both VRCP–I and VRCP–C is that they guarantee that they can only increase the size of the prediction set for any input $\boldsymbol{x}$ compared to vanilla CP, thus will always attain at least as much coverage as the vanilla CP procedure. Moreover, as we show in Section 5, both algorithms do not trivially inflate the size of the prediction sets and maintain a similar distribution of set sizes. This is formalised in the Proposition 1, which is proved in Appendix A.

**Proposition 1.** *Let $C(\boldsymbol{x})$ and $C_\epsilon(\boldsymbol{x})$ be the prediction sets obtained using vanilla CP and VRCP (using VRCP–I or VRCP–C), respectively. For any input $\boldsymbol{x}$, we have that $C(\boldsymbol{x}) \subseteq C_\epsilon(\boldsymbol{x})$.*

## 5  Evaluation

We evaluate VRCP–I and VRCP–C on classification (image) and regression (RL) benchmarks, and compare them against the SotA approaches on each benchmark. For all the networks used, we did not perform adversarial training as we assume that the attack budget $\epsilon$ is unknown at training time. Nonetheless, both our approaches can benefit from adversarial training, as it results in models that are more verifiable and have tighter bounds for the same attack budget.[3]

---

[3]Code for the experiments is available at: https://github.com/ddv-lab/Verifiably_Robust_CP

## 5.1 Classification Experiments

We evaluate each method using a nominal coverage of $1 - \alpha = 0.9$ and report the 95% confidence intervals for coverage and average set sizes computed over 50 splits ($n_{\text{splits}} = 50$) of the calibration, holdout and test set.

**Bounds** We use the verification library auto_LiRPA (Xu et al., 2020a) to compute the output bounds for $f(\boldsymbol{x})$ required in Eq. (10) and Eq. (11) for VRCP–I and VRCP–C respectively. In particular, we use two SotA GPU-parallelised incomplete NN verification algorithms, CROWN Zhang et al. (2018b) and $\alpha$-CROWN Xu et al. (2020b). In brief, CROWN performs linear bound propagation and $\alpha$-CROWN employs a branch-and-bound algorithm to tighten the CROWN bounds at the expense of slower verification times. Therefore, we use CROWN to compute the output bounds for the TinyImageNet model and $\alpha$-CROWN for the smaller CIFAR10 and CIFAR100 models.

Our CIFAR10 model with $\alpha$-CROWN takes $\approx 0.5$s per image to compute bounds with $\epsilon = 0.03$, whereas our larger CIFAR100 model takes $\approx 7.2$s with $\epsilon = 0.02$. Comparatively, computing the smoothed scores takes $\approx 0.09$s per image to compute on both models under the same respective $\epsilon$ values. The largest model for the TinyImageNet dataset uses CROWN to compute bounds at a rate of $\approx 0.2$s per image whereas the smoothed scores take $\approx 0.24$s. All measurements are made with respect to the hardware details listed in Appendix B.

**Attacks** We use the PGD attack algorithm (Madry et al., 2017), which is a popular white-box attack algorithm to generate adversarial inputs with respect to either the $\ell_2$ or $\ell_\infty$-norm.

**Models** For all datasets, we train a CNN model on training set images with random crop and horizontal flip augmentations. Full model details are outlined in the appendix.

**Hyperparameters** RSCP+ based approaches use $\sigma = 2\epsilon$, $\beta = 0.001$ and those with PTT use $|\mathcal{D}_{\text{hold}}| = 500$, $b = 0.9$ and $T = 1/400$. For PGD, we choose a step size of $1/255$ and compute 100 steps for each attack. For CIFAR10 and CIFAR100 $|\mathcal{D}_{\text{train}}| = 50{,}000$ and for TinyImageNet $|\mathcal{D}_{\text{train}}| = 100{,}000$. For all datasets $|\mathcal{D}_{\text{cal}}| = 4{,}500$ and $|\mathcal{D}_{\text{test}}| = 5{,}000$.

**Results** In Table 1, we benchmark both our methods against the baseline vanilla CP (which is agnostic of the attack), RSCP+ and RSCP+ with PTT. At inference time, images are attacked using PGD to generate $\ell_2$-norm bounded attacks with $\epsilon = 0.02$ for CIFAR100 and TinyImageNet, and $\epsilon = 0.03$ for CIFAR10.

In all domains, the vanilla CP method fails to construct valid prediction sets with nominal marginal coverage, as expected. RSCP+ maintains robust marginal coverage but produces trivial prediction sets in all settings due to the highly conservative inflation of the threshold with respect to the calibration scores. Using PTT improves RSCP+'s performance but introduces significant variance in the set sizes: in many cases, PTT still produces trivial prediction sets and is heavily dependent on the sampled holdout set for RSCP+ to generate useful predictions.

Both of our methods have minimal sample dependence, as demonstrated by a very small variability in coverage and size over the 50 splits. We obtain prediction sets with substantially smaller average sizes than the other robust approaches, and hence, they provide more informative uncertainty estimates. VRCP–I provides slightly more efficient regions than VRCP–C. Still, it implies additional computational overhead at inference time because it requires computing bounds via NN verification for each test sample. In contrast, in VRCP–C, bounds are computed only once at calibration time. On the other hand, in an environment where we may want to change $\epsilon$ for different test points at inference time, VRCP–I would be a sound choice, while VRCP–C would require re-calibration.

**Effect of increasing adversarial noise** Fig. 2b shows the impact of increasing $\epsilon$ across all evaluated robust methods. Our methods consistently produce smaller average set sizes with minor sample dependence, and simultaneously provide a more conservative marginal coverage than RSCP+ (PTT). We remark that, unlike RSCP+, we do not require a holdout set or any score function transformations.

**Effect of increasing Monte-Carlo samples** Fig. 2a displays the influence of the $n_{\text{MC}}$ hyperparameter on the RSCP+ based methods with respect to our CIFAR10 model. Whilst increasing samples

Table 1: Marginal Coverage and Average Set Sizes for different methods on CIFAR10, CIFAR100 and TinyImageNet. All results record a 95% confidence interval with $n_{\text{splits}} = 50$, $\alpha = 0.1$, $\sigma = 2\epsilon$, $n_{\text{MC}} = 1024$, $\epsilon = 0.03$ for CIFAR10 and $\epsilon = 0.02$ otherwise.

| | CIFAR10 | | CIFAR100 | | TinyImageNet | |
|---|---|---|---|---|---|---|
| Method | Coverage | Size | Coverage | Size | Coverage | Size |
| Vanilla | 0.878±0.002 | 1.721±0.008 | 0.890±0.002 | 6.702±0.058 | 0.886±0.002 | 38.200±0.252 |
| RSCP+ | 1.000±0.000 | 10.000±0.000 | 1.000±0.000 | 100.000±0.000 | 1.000±0.000 | 200.000±0.000 |
| RSCP+ (PTT) | 0.983±0.008 | 8.357±0.780 | 0.925±0.010 | 26.375±9.675 | 0.931±0.013 | 90.644±20.063 |
| VRCP–I | 0.986±0.000 | **4.451±0.011** | 0.971±0.001 | **22.530±0.107** | 0.958±0.001 | **72.486±0.311** |
| VRCP–C | 0.995±0.000 | 5.021±0.010 | 0.983±0.000 | 23.676±0.131 | 0.965±0.001 | 77.761±0.352 |

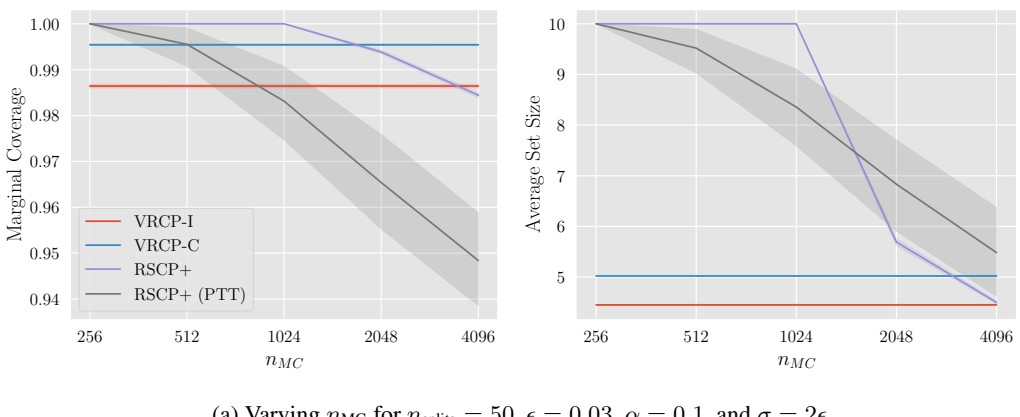

(a) Varying $n_{\text{MC}}$ for $n_{\text{splits}} = 50$, $\epsilon = 0.03$, $\alpha = 0.1$, and $\sigma = 2\epsilon$.

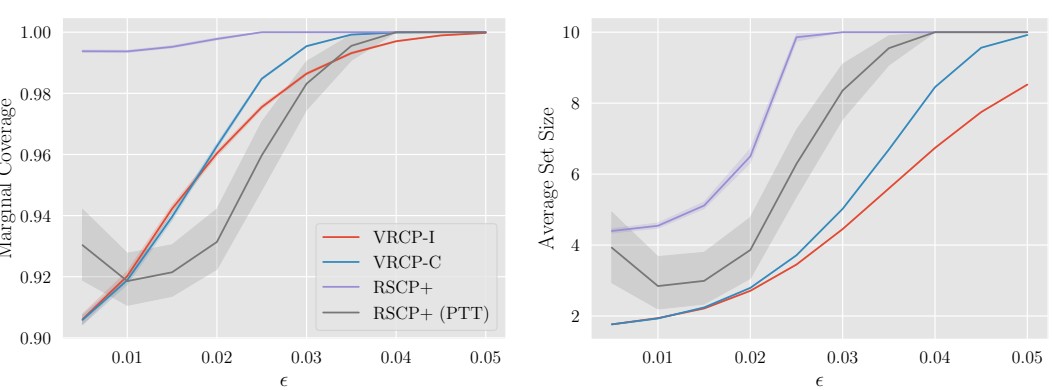

(b) Varying the value of $\epsilon$ for $n_{\text{splits}} = 50$, $\sigma = 2\epsilon$, $\alpha = 0.1$, and $n_{\text{MC}} = 1024$.

Figure 2: Marginal Coverage and Average Set Sizes on CIFAR100 with 95% confidence intervals.

improves the performances of randomised smoothing approaches, we incur a large computational overhead when computing the smoothed scores. In our experiments in Table 1 we fix $n_{\text{MC}} = 1024$ which is four times larger than the value for $n_{\text{MC}}$ used in previous work (Gendler et al., 2021; Yan et al., 2023) as a trade-off between prediction quality and computation.

**Beyond $\ell_2$-norm bounded attacks**  Table 2 demonstrates that both of our methods generalise to other $\ell_p$-bounded perturbations other than for when $p = 2$ which RSCP+ is limited to. In particular, we examine the $\ell_\infty$, where even a small $\epsilon$ can cause misclassification. We experiment using CIFAR10 and use $\epsilon = 0.001$. PGD is used to generate $\ell_\infty$-bounded adversarial examples.

**Set size distribution**  From Fig. 1 we can visually examine the sample dependency issue that the PTT introduces. In the splits where the holdout set allows the PTT to make an informative transformation, RSCP+ is able to make quite reasonable predictions, otherwise, RSCP+ just returns trivial sets. This is clearly an undesirable property and adds significant variance to the predictions.

Table 2: Marginal Coverage and Average Set Sizes for $\epsilon$ perturbations with respect to the $\ell_\infty$-norm on the CIFAR10 dataset. All results record a 95% confidence interval with $n_{\text{splits}} = 50$, $\alpha = 0.1$ and $\epsilon = 0.001$.

| | CIFAR10 | |
|---|---|---|
| Method | Coverage | Size |
| Vanilla | 0.872±0.002 | 1.737±0.007 |
| VRCP–I | 0.947±0.001 | **2.262±0.008** |
| VRCP–C | 0.931±0.001 | 2.342±0.008 |

Both of our methods increase the spread of the average set sizes to account for the presence of adversarial examples whilst still maintaining a consistent distribution.

## 5.2 Regression Experiments

We evaluate our VRCP framework on regression tasks from the PettingZoo Multi-Particle Environment (MPE) library Terry et al. (2021) for deep reinforcement learning. In these environments, the world is a 2D space containing $n$ agents (of which some may be adversarial) and $m$ landmarks, which are defined as circles of fixed radii. The position of the landmarks is fixed, and agents traverse the space according to second-order motion laws. We evaluate our method on three tasks:

- **Adversary** The good agents must try to reach a specific goal landmark whilst avoiding the adversaries. We use 2 good agents, 1 adversary and 2 landmarks.

- **Spread** All agents collaborate and minimise the distance to each landmark. We set the number of agents and landmarks equal to 3.

- **Push** In this task, there is a single good agent, adversary and landmark. The task is for the adversary to maximise the distance between the landmark and the good agent.

In our experiments, for data-generation we select 5,000 random initial world configurations and, for each, simulate 25 Monte-Carlo trajectories of length $k = 5$. The regression task for all environments is to predict the upper and lower quantiles of the total cumulative reward over the $k$ steps, given as input the initial world state. As in the classification experiments, we partition the dataset into the following partitions: $|\mathcal{D}_{\text{train}}| = 1,000$, $|\mathcal{D}_{\text{cal}}| = 2,000$ and $|\mathcal{D}_{\text{test}}| = 2,000$.

For computing the bounds, we use CROWN Zhang et al. (2018b) with $\ell_\infty$-bounded perturbations. To generate the adversarially perturbed test points, we use the Fast Gradient Sign Method as given in (Goodfellow et al., 2015).

As seen in Table 3, both VRCP methods recover the marginal coverage guarantees in the presence of adversarial perturbations, whereas vanilla CP fails drastically after $\epsilon = 0.02$. We note that the performance of VRCP–C and VRCP–I are similar, although VRCP–I tends to produce more conservative intervals (without sacrificing efficiency).

## 6   Limitations

VRCP's scalability depends on that of the underlying neural network verifier. We evaluated VRCP on small to medium-sized neural networks. For large networks, existing complete verification methods

Table 3: Marginal coverage and average interval lengths for each MPE regression task for various $\epsilon$ perturbations bounded by an $\ell_\infty$-norm. All results record a 95% confidence interval with $n_{\text{splits}} = 50$.

| | Perturbation | $\epsilon = 0.01$ | | $\epsilon = 0.02$ | | $\epsilon = 0.04$ | |
|---|---|---|---|---|---|---|---|
| | Method | Coverage | Length | Coverage | Length | Coverage | Length |
| **Adversary** | Vanilla | 0.871±0.006 | 0.480±0.006 | 0.834±0.007 | 0.484±0.006 | 0.745±0.009 | 0.490±0.006 |
| | VRCP–I | 0.928±0.004 | 0.605±0.006 | 0.951±0.003 | 0.673±0.006 | 0.985±0.002 | 0.855±0.006 |
| | VRCP–C | 0.910±0.005 | 0.534±0.006 | 0.923±0.005 | 0.606±0.006 | 0.966±0.003 | 0.806±0.005 |
| **Spread** | Vanilla | 0.864±0.005 | 0.595±0.005 | 0.834±0.005 | 0.602±0.005 | 0.768±0.006 | 0.612±0.005 |
| | VRCP–I | 0.929±0.004 | 0.690±0.006 | 0.958±0.003 | 0.769±0.006 | 0.991±0.001 | 0.992±0.006 |
| | VRCP–C | 0.908±0.005 | 0.663±0.006 | 0.935±0.004 | 0.762±0.005 | 0.977±0.002 | 1.054±0.006 |
| **Push** | Vanilla | 0.891±0.006 | 0.643±0.006 | 0.875±0.007 | 0.646±0.006 | 0.841±0.008 | 0.652±0.006 |
| | VRCP–I | 0.917±0.006 | 0.687±0.006 | 0.934±0.005 | 0.721±0.006 | 0.961±0.003 | 0.800±0.006 |
| | VRCP–C | 0.905±0.005 | 0.674±0.006 | 0.910±0.005 | 0.711±0.005 | 0.924±0.005 | 0.795±0.005 |

would become computationally infeasible, while incomplete methods would produce bounds that are too loose to be useful. However, it is important to note that since VRCP is agnostic of the specific verification tool used, it would directly benefit from any future advances in neural network verification. Thus, as neural network verification tools continue to evolve and improve, so does VRCP.

## 7 Conclusion

We introduced Verifiably Robust Conformal Prediction (VRCP), a novel framework that leverages conformal prediction and neural network verification to produce prediction sets that maintain marginal coverage under adversarial perturbations. We presented two variants: VRCP–C, which applies verification at calibration time, and VRCP–I, which applies verification at inference time.

Extensive experiments on classification and regression tasks demonstrated that VRCP recovers valid marginal coverage in the presence of $\ell_1$, $\ell_2$, and $\ell_\infty$-norm bounded adversarial attacks while producing more accurate prediction sets than existing methods. VRCP is the first adversarially robust CP framework supporting regression tasks and perturbations beyond the $\ell_2$-norm, achieving strong results without relying on probabilistic smoothing or posthoc corrections. VRCP's theoretical guarantees and empirical performance showcase the potential of leveraging verification tools for uncertainty quantification of machine learning models under attack.

## Acknowledgments and Disclosure of Funding

This work is supported by the "REXASI-PRO" H-EU project, call HORIZON-CL4-2021-HUMAN-01-01, Grant agreement ID: 101070028.

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

# A  Additional Proof Details

Here we prove Proposition 1 regarding the prediction sets obtained from VRCP–I and VRCP–C.

*Proof of Proposition 1.* To prove $C(\boldsymbol{x}) \subseteq C_\epsilon(\boldsymbol{x})$ for VRCP–I it suffices to observe that

$$C_\epsilon(\boldsymbol{x}) = \{y \in Y : s^\perp(\boldsymbol{x}, y) \le Q_{1-\alpha}(F)\} \supseteq \bigcup_{\boldsymbol{x}' \in B_\epsilon(\boldsymbol{x}_{n+1})} \{y \in Y : S(\boldsymbol{x}', y) \le Q_{1-\alpha}(F)\}$$

$$= \bigcup_{\boldsymbol{x}' \in B_\epsilon(\boldsymbol{x}_{n+1})} C(\boldsymbol{x}') \supseteq C(\boldsymbol{x}).$$

To prove the same for VRCP–C, we observe that since $Q_{1-\alpha}(F^\top) \ge Q_{1-\alpha}(F)$, we have that

$$C_\epsilon(\boldsymbol{x}) = \{y \in Y : S(\boldsymbol{x}, y) \le Q_{1-\alpha}(F^\top)\} \supseteq \{y \in Y : S(\boldsymbol{x}, y) \le Q_{1-\alpha}(F)\} = C(\boldsymbol{x}). \qquad \square$$

Next, we prove the validity of the VRCP–I region for the regression case, defined in Eq. (15).

*Proof.* It suffices to show that all $y \in C_\epsilon(\boldsymbol{x}_{n+1}) = [f^\perp - q, f^\top + q]$ satisfy $s^\perp(\boldsymbol{x}_{n+1}, y) = \max\{f^\perp - y, y - f^\top\} \le q$ and all $y \notin C_\epsilon(\boldsymbol{x}_{n+1})$ do not. For simplicity of notation, we abbreviated $f^\top_{\alpha_{\text{high}}}(\boldsymbol{x}_{n+1})$ with $f^\top$, $f^\perp_{\alpha_{\text{low}}}(\boldsymbol{x}_{n+1})$ with $f^\perp$ and $Q_{1-\alpha}(F)$ with $q$.

Assume $y \in C_\epsilon(\boldsymbol{x}_{n+1})$. We divide the proof into two cases:

1. $s^\perp(\boldsymbol{x}_{n+1}, y) = f^\perp - y$, which implies that $y \in [f^\perp - q, \frac{f^\top + f^\perp}{2}]$. It suffices to show that $f^\perp - y \le q$ for $y = f^\perp - q$, which is clearly satisfied.

2. $s^\perp(\boldsymbol{x}_{n+1}, y) = y - f^\top$, which implies that $y \in [\frac{f^\top + f^\perp}{2}, f^\top + q]$. It suffices to show that $y - f^\top \le q$ for $y = f^\top + q$, which is clearly satisfied.

Finally, we show that $y \notin C_\epsilon(\boldsymbol{x}_{n+1})$ implies $s^\perp(\boldsymbol{x}_{n+1}, y) > q$: if $y < f^\perp - q$, we have that $s^\perp(\boldsymbol{x}_{n+1}, y) = f^\perp - y > q$. Similarly, if $y > f^\perp + q$, we have that $s^\perp(\boldsymbol{x}_{n+1}, y) = y - f^\top > q$. $\qquad \square$

# B  Model Details

All experimental results were obtained from running the code provided in our GitHub repository on a server with 2x Intel Xeon Platinum 8360Y (36 cores, 72 threads, 2.4GHz), 512GB of RAM and an NVIDIA A40 48GB GPU. All pre-trained models as well as the training scripts are also provided in the GitHub repository. In summary, the models' train and test performances are provided in Tables 4 and 5.

Table 4: Train and test accuracies (%) for the classifications models on CIFAR10, CIFAR100, and TinyImageNet datasets. It should be noted that the model's accuracy has no effect on VRCP's validity and only affects the efficiency of the prediction sets (more accurate models, tighter prediction regions)

| Metric | CIFAR10 | CIFAR100 | TinyImageNet |
|---|---|---|---|
| Train Top-5 | 98.77 | 90.49 | 78.44 |
| Train Top-1 | 77.80 | 67.12 | 52.81 |
| Test Top-5 | 98.27 | 82.87 | 55.72 |
| Test Top-1 | 76.52 | 55.73 | 29.65 |

## B.1  Classification

**CIFAR10**  We use 2 convolution layers with average pooling and dropout, followed by 2 linear layers. ReLU activations across all layers.

**CIFAR100**  We use 1 convolution layer with average pooling, 2 further convolution layers with average pooling and dropout followed by 2 linear layers. ReLU activations across all layers.

**TinyImageNet**  We use 4 convolution layers with dropout followed by 2 linear layers with dropout. Leaky ReLU activation function with $a = 0.1$

For all models we train using images augmented with random crop with 4 pixels of padding and random horizontal flip. We standardise the TinyImageNet models with $\mu = 0.5$ and $\sigma = 0.5$ overall 3 RGB channels.

As previously mentioned, we do not make any assumptions during training about the perturbations we expect to see at inference time. As such, unlike the existing SotA methods, we do not train on smoothed or adversarially attacked images.

All models are trained for 200 epochs with a batch size of 128 using the stochastic gradient descent optimiser with momentum set to 0.9. We also employ a weight decay of $5 \times 10^{-4}$ and a cosine annealing learning rate scheduler.

## B.2   Regression

For the MPE datasets, we train Deep Q-Net policies for the RL tasks for the sole purposes of generating the appropriate datasets and provide these policies in the GitHub repository.

The model used for the quantile regressors is a simple linear architecture consisting of 3 layers, separated with ReLU activation functions and dropout. We trained the model to estimate the $\alpha/2$ and $1 - \alpha/2$ quantiles, where $\alpha = 0.1$, as in the other experiments.

The exact parameters for the RL policies can be found in the config files within the GitHub repository, however have little bearing on the efficiency of our results, being used only for the data-generating process. The quantile regressors are each trained for 400 epochs, with a learning rate of $10^{-5}$, dropout of 0.1 and a decay of $10^{-5}$.

Table 5: Train and test loss for the regression models in the adversary, spread, and push environments.

| Metric | Adversary | Spread | Push |
|--------|-----------|--------|------|
| Train  | 0.066     | 0.075  | 0.075 |
| Test   | 0.051     | 0.053  | 0.068 |

