# OpenReview forum: "Verifiably Robust Conformal Prediction"
_NeurIPS.cc/2024/Conference — NeurIPS 2024 poster_

### Official Review · Reviewer_gWvg · 2024-07-05

**Soundness:** 3
**Presentation:** 3
**Contribution:** 2
**Rating:** 5
**Confidence:** 3

**Summary:**

The authors propose a novel method to recover coverage guarantees for conformal predictions in the presence of adversarial attacks. Unlike previously proposed approaches, the authors directly leverage verifiable methods for NN to compute prediction scores. Through empirical tests, the authors prove the benefit of the proposed approach against attacks bounded by different norms.

**Strengths:**

1. Empirical results support the benefit of the proposed approach against vanilla CP, RSCP+

2. The proposed approach seems to be robust across different values of the adversarial perturbation

3. The approach is well-described and the coverage guarantees are supported by theoretical arguments

**Weaknesses:**

1. In the paper, the authors focus on PDG. However, the literature offers more advanced and sophisticated attacks. It would be beneficial to assess the robustness of the proposed approach against other type of attacks.

2. Since it is not specified in the paper, I assume the ML model under attack has not been trained using adversarial training. The reader might wonder whether the benefit of the proposed approach remains valid in the presence of an adversarially trained model. Concerning the latter case, will vanilla CP still violate the coverage guarantees?

3. Figure 2 is hard to read. Which curve represents the proposed approach, and which one represents the benchmarks?
Concerning Figure 2a: Why does the coverage go to one as the magnitude of the adversarial perturbation goes to one? Perhaps it is because all the methods converge to a trivial prediction set. However, this should be clearly explained in the text.
Concerning Figure 2b: Why does the coverage decrease as a function of the number of samples?

**Questions:**

See Weaknesses

**Limitations:**

Limitations are discussed

---

> ### Author Rebuttal · Authors · 2024-08-07
>
> > 1. In the paper, the authors focus on PDG. However, the literature offers more advanced and sophisticated attacks. It would be beneficial to assess the robustness of the proposed approach against other type of attacks.
>
> With regard to the chosen attack methods, we evaluate PGD in the case of classification tasks and FGSM for the regression task. We chose PGD as it remains a popular choice within existing literature on robust CP approaches, and so as to be consistent and draw fair comparisons against the RSCP/+ methods. In theory, however, our approach is agnostic to the adversarial attack algorithm used to perturb the inputs, as it ensures robustness to any $\ell^p$ bounded adversarial attacks.
>
> > 2. Since it is not specified in the paper, I assume the ML model under attack has not been trained using adversarial training. The reader might wonder whether the benefit of the proposed approach remains valid in the presence of an adversarially trained model. Concerning the latter case, will vanilla CP still violate the coverage guarantees?
>
> We appreciate that we should have been clearer about the adversarial training aspect, and your assumption is correct in that we do not use adversarial training. Whilst an adversarially trained model is more reliable on adversarial inputs, using one in combination with vanilla CP doesn't guarantee robust coverage. For this to hold, we would require the adversarially trained model to guarantee prediction invariance for every input and the corresponding $\epsilon$-bounded region around it, which is not the case. Our VRCP method would provide valid prediction regions instead. We will clarify this point further in the paper.
>
> > 3. Figure 2 is hard to read. Which curve represents the proposed approach, and which one represents the benchmarks? Concerning Figure 2a: Why does the coverage go to one as the magnitude of the adversarial perturbation goes to one? Perhaps it is because all the methods converge to a trivial prediction set. However, this should be clearly explained in the text. Concerning Figure 2b: Why does the coverage decrease as a function of the number of samples?
>
> Perhaps confusingly, the legend for all the plots can be seen in the 3rd figure due to an oversight in figure placement. We will update the manuscript to include the legend more clearly below the figures.
>
> As the reviewer pointed out, in Figure 2a, coverage and prediction set size increase for larger epsilon due to the robust methods including more classes within their prediction sets to account for the stronger potential perturbation. This trend converges to trivial prediction sets and a coverage of 1. Note that in our experiments with $\epsilon=0.05$, VRCP-I has an empirical coverage of $1$ but does not produce trivial prediction sets (its average set size is $8.52$ against a maximum of $10$).
>
> In Figure 2b, we observe that increasing the number of Monte-Carlo samples used in RSCP+ methods improves their prediction set efficiency. The higher the number of samples used for randomised smoothing, the lower the effect is of the Hoeffding/Bernstein bound required to correct for sampling error.
>
> We will add both of these discussions to the main text to improve clarity.

---

> > ### Comment · Reviewer_gWvg · 2024-08-12
> >
> > I would like to thank the authors to take the time to write the rebuttal to address my concerns.
> >
> > Although I appreciate the paper motivation, I think that the contribution is limited. I will keep my previous score.

---

### Official Review · Reviewer_iDUp · 2024-07-12

**Soundness:** 3
**Presentation:** 3
**Contribution:** 2
**Rating:** 5
**Confidence:** 4

**Summary:**

The paper provides a verifiably robust conformal prediction method via neural network verification. It considers two paradigms for considering the perturbation either at the calibration stage or the inference stage. They also consider the validity of method for both classification and regression.

**Strengths:**

1. The paper is well written. I believe people with other background can also understand it well.
2. The methods work for me. It is clear that NN verification provides a lower/upper bound of the logits, which are transformed to certified bound for conformity scores.

**Weaknesses:**

1. Novelty: I understand that it is a new angle that combines NN verification and conformal prediction to provide a certifiably robust conformal prediction framework, but I wonder technically, can we do more about the combination. For example, can we have new conformity score adapted to NN verification that can provide tighter bound? Or at least empirically, we can analyze what types of conformity scores are more suitable for randomized certification? What are more suitable for deterministic certification (NN verification).

2. What is the intuition that the method can outperform RSCP+ in Table 1? Since usually randomized smoothing provides better certification than NN verification, then coming to the conformal prediction context, what makes the difference?

3. Missing related work: Kang, Mintong, et al. COLEP: Certifiably Robust Learning-Reasoning Conformal Prediction via Probabilistic Circuits. ICLR 2024.

**Questions:**

Please refer to the weakness part. Overall, I do not have many concerns about the paper's soundness. If the authors can provide more insights into comparisons between randomized certification and deterministic certification in robust conformal prediction, it would be beneficial to the community. Basically, the authors need more support of why NN verification-based conformal prediction is worthy besides it can provide a general $\ell_p$ norm guarantee.

**Limitations:**

discussed in Sec 6.1

---

> ### Author Rebuttal · Authors · 2024-08-07
>
> > 1. Novelty: I understand that it is a new angle that combines NN verification and conformal prediction to provide a certifiably robust conformal prediction framework, but I wonder technically, can we do more about the combination. For example, can we have new conformity score adapted to NN verification that can provide tighter bound? Or at least empirically, we can analyze what types of conformity scores are more suitable for randomized certification? What are more suitable for deterministic certification (NN verification).
>
> We address the question regarding verification-friendly non-conformity score functions in **Part 1** of the **Global Rebuttal**.
>
> > 2. What is the intuition that the method can outperform RSCP+ in Table 1? Since usually randomized smoothing provides better certification than NN verification, then coming to the conformal prediction context, what makes the difference?
>
> We address the question regarding the intuition behind VRCP’s improved empirical performance to RSCP+ in **Part 2** of the **Global Rebuttal**.
>
> > 3. Missing related work: Kang, Mintong, et al. COLEP: Certifiably Robust Learning-Reasoning Conformal Prediction via Probabilistic Circuits. ICLR 2024.
>
> We will add this to the related work of the updated manuscript. We understand that this paper introduces a learning-reasoning framework (COLEP) that integrates the original RSCP (Gendler et al. 2021) to provide adversarially robust conformal prediction sets by using auxiliary models and probabilistic circuits. This work is complementary to ours because VRCP can be directly applied to COLEP in place of RSCP.
>
> > Please refer to the weakness part. Overall, I do not have many concerns about the paper's soundness. If the authors can provide more insights into comparisons between randomized certification and deterministic certification in robust conformal prediction, it would be beneficial to the community. Basically, the authors need more support of why NN verification-based conformal prediction is worthy besides it can provide a general $\ell_p$ norm guarantee.
>
> We addressed this question in detail above. We would also like to note that VRCP provides more benefits as listed below:
>
> - Other existing robust CP approaches introduce additional dependencies, such as the need for a hold-out set or a large number of samples (used for randomized smoothing)
> - VRCP supports regression tasks without requiring any modifications, unlike the existing approaches
> - VRCP supports arbitrary $\ell^p$ norms

---

> > ### Comment · Reviewer_iDUp · 2024-08-11
> > **Thanks for the rebuttal**
> >
> > Thank the authors for the rebuttal. Overall, I think this paper is technically sound and the first combination work of NN verification and conformal certification. I will maintain a borderline because the motivation for using it instead of RSCP is not very strong besides the arbitrary $\ell_p$ norm. Although, RSCP requires many samples, this work is also not efficient due to NN verification. I also do not think RSCP can not be tailored for regression tasks.

---

### Official Review · Reviewer_cbGx · 2024-07-16

**Soundness:** 3
**Presentation:** 4
**Contribution:** 3
**Rating:** 6
**Confidence:** 4

**Summary:**

This submission proposes VRCP, a framework for verifiably robust conformal prediction under Lp-bounded adversarial attacks. The framework integrates existing bound propagation tools for verification of conformal predictions. Two variants, VRCP-I that triggers bound computation at inference time and VRCP-C that triggers bound computation at calibration time, are proposed. VRCP-I has inference overhead but the set size is smaller. Experiments on CIFAR10, CIFAR100, and a regression task demonstrate the effectiveness.

**Strengths:**

1. The submission introduces a feasible method to turn well-studied neural network verifiers into conformal prediction verifiers. The method is sound in theory and is also verified empirically. This is a novel contribution to the field.

2. Compared to existing verifiably robust conformal prediction methods, VRCP has superior empirical performance especially when compared to randomized smoothing, demonstrating its practical value.

3. Presentation is generally great and easy to follow.

**Weaknesses:**

1. The framework is relatively straightforward. The foundation is some probability relaxations that can be intuitively derived. I would not view this intuitiveness as a weakness. However, it would be great if the authors could discuss some extensions and optimizations, e.g., proof sharing for VRCP-C, alternative but more verification-friendly score function design, etc.

2. Some in-depth discussion could benefit the submission. Concretely, why does VRCP work with different Lp norms? Does the benefit come from the existing verifier's flexibility? Does the bound tightness differ with different Lp norms? Why does the method surpass RSCP?

3. The experimental setup is not very clear: What is the model's normal accuracy? Could you provide more background on the setup of regression experiments especially the physical meaning of the reward bound and nominal performance? The setting is not very common in the literature.

Minor:
Line 129-130: the lengthy sentence seems to be lacking a period or comma.

**Questions:**

See weaknesses.

**Limitations:**

Yes, authors discuss the limitations in the last paragraph of the main text.

---

> ### Author Rebuttal · Authors · 2024-08-07
>
> > 1. The framework is relatively straightforward. The foundation is some probability relaxations that can be intuitively derived. I would not view this intuitiveness as a weakness. However, it would be great if the authors could discuss some extensions and optimizations, e.g., proof sharing for VRCP-C, alternative but more verification-friendly score function design, etc.
>
> As you correctly mention, our methods rely on the output bounds computed by the NN verifier. Thus, the same features of NN architectures that are friendly for verification will be beneficial to our methods. For example, using transformations and activation functions with low Lipschitz constants would result in tighter bounds with linear bound propagation approaches.
> We address the question regarding verification-friendly non-conformity score functions in (1) of the global review.
>
> > 2. Some in-depth discussion could benefit the submission. Concretely, why does VRCP work with different Lp norms? Does the benefit come from the existing verifier's flexibility? Does the bound tightness differ with different Lp norms? Why does the method surpass RSCP?
>
> Regarding the first two questions, the existing verification methods indeed grant VRCP's ability to extend to other $\ell^p$ norms. These methods theoretically provide verification for perturbations bounded by any $\ell^p$ norm, although in practice, often only the most common $\ell^p$ norms are implemented ($\ell^1, \ell^2$ and $\ell^{\infty}$).
>
> The bounds' tightness differs with different $\ell^p$ norms because, for the same epsilon bound, the epsilon-bounded $\ell^1$ ball is strictly smaller (in volume) than the corresponding $\ell^2$ ball, which is in turn smaller than the $\ell^{\infty}$ ball. Hence, for larger values of $p$, we will see larger input regions and looser output bounds.
>
> We address the question regarding the intuition behind VRCP’s improved empirical performance to RSCP+ in (2) of the global review.
>
> > 3. The experimental setup is not very clear: What is the model's normal accuracy? Could you provide more background on the setup of regression experiments especially the physical meaning of the reward bound and nominal performance? The setting is not very common in the literature.
>
> For the classification experiments, the test accuracies of the models are:
>
> |Accuracy/Model|CIFAR10|CIFAR100|TinyImageNet|
> |---|---|---|---|
> |Top-5 Test|98.27%|82.87%|55.72%|
> |Top-1 Test|76.52%|55.73%|29.64%|
>
> It should be noted that the accuracy of the model has no effect on VRCP's validity and only affects the efficiency of the prediction sets (more accurate models, tighter prediction regions).
>
> For the regression experiments, the train and test losses of the models are:
> |Loss/Environment|Adversary|Spread|Push|
> |---|---|---|---|
> |Train|0.066|0.075|0.075|
> |Test|0.051|0.053|0.068|
>
> In terms of the reward bound, we have scaled the total cumulative reward between the range [0, 1] and thus the prediction intervals are also taken over this range. We will clarify this in the updated paper.

---

> > ### Comment · Reviewer_cbGx · 2024-08-14
> >
> > Thanks for the response! Most of my concerns are resolved. So I maintain my score.
> >
> > >  Does the bound tightness differ with different Lp norms?
> >
> > Sorry the question may not be clear enough. I was asking whether the gap between verifier's score bound and actual score minimum / maximum for perturbed inputs can become larger/smaller among different Lp norms.

---

> > > ### Author Response · Authors · 2024-08-14
> > >
> > > Thank you for your response.
> > >
> > > > Sorry the question may not be clear enough. I was asking whether the gap between verifier's score bound and actual score minimum / maximum for perturbed inputs can become larger/smaller among different Lp norms.
> > >
> > > Yes, it differs between different $\ell_p$ norms. For example, $\ell_\infty$ results in larger gaps for the same attack budget $\epsilon$. This is because the $\ell_\infty$ ball contains other $\ell_p$ balls. However, it should be noted that verifiers tend to over-approximate non-linear norms (e.g., $\ell_2, \ell_3, \dots$) more than linear norms (e.g., $\ell_1$ and $\ell_\infty$). This means for non-linear norms, we may get larger gaps than linear norms.
> > >
> > > We hope this answers your questions.

---

### Author Rebuttal · Authors · 2024-08-07

We thank the reviewers for their useful comments.

Here we respond to the common feedback amongst all the reviews.

---

## Part 1. Verification-Friendly Non-Conformity Score Functions

We agree that investigating verification-friendly score functions is a great idea for future work. We use $1-f_y(x)$ as the score function, as we find this to be, for classification tasks, the most popular choice across the literature, including existing randomised smoothing approaches. This function happens to be verification-friendly because it is completely linear and does not introduce further over-approximations. Score functions that introduce over-approximation (e.g., where there are multiple expressions to bound) would make our approach more conservative and possibly favour randomised smoothing approaches.

---

## Part 2. The intuition behind VRCP's Improved Empirical Performance to RSCP+

In the RSCP+ approach, the robustness guarantee is dependent on a number of factors that affect performance, listed below:

- The size of the Lipschitz constant $\sigma/\epsilon$.
- The size of the Hoeffding/Bernstein bounds used to correct for sampling error when estimating the mean of their smoothed scores.
- The representativity of the holdout set w.r.t. the rest of the calibration distribution (when using PTT).

Both VRCP-C and VRCP-I methods perform agnostically of the aforementioned factors and thus exhibit improved efficiency where RSCP/+ falls short.

---

### Decision · Program_Chairs · 2024-09-25

**Decision:**

Accept (poster)

**Comment:**

This paper considers the problem of providing valid prediction sets using conformal prediction in the presence of adversarial perturbations. This is an active research area and the paper makes a good contribution by approaching this problem through the lens of neural network verification tools and is applicable to both classification and regression tasks (demonstrated by experiments).

The reviewers' appreciated the new ideas in the paper but also raised some good questions. Authors' responses were satisfactory to address those questions.

Therefore, I recommend accepting the paper and strongly encourage the authors' to incorporate all the discussion in the camera copy to further improve the paper.